# Extracellular ATP Contributes to Barrier Function and Inflammation in Atopic Dermatitis: Potential for Topical Treatment of Atopic Dermatitis by Targeting Extracellular ATP

**DOI:** 10.3390/ijms252212294

**Published:** 2024-11-15

**Authors:** Kazuhiko Yamamura, Fumitaka Ohno, Shu Yotsumoto, Yuki Sato, Nanae Kimura, Kiichiro Nishio, Keiichi Inoue, Toshio Ichiki, Yoko Kuba-Fuyuno, Kei Fujishima, Takamichi Ito, Makiko Kido-Nakahara, Gaku Tsuji, Takeshi Nakahara

**Affiliations:** 1Department of Dermatology, Graduate School of Medical Sciences, Kyushu University, Fukuoka 812-8582, Japan; yamamura.kazuhiko.821@m.kyushu-u.ac.jp (K.Y.);; 2Research and Clinical Center for Yusho and Dioxin, Kyushu University Hospital, Fukuoka 812-8582, Japan

**Keywords:** atopic dermatitis, adenosine triphosphate, filaggrin, TEWL, barrier function, TARC

## Abstract

Atopic dermatitis (AD) is characterized by chronic inflammation, barrier dysfunction, and pruritus, exacerbated by external stimuli, such as scratching. This study investigates the role of extracellular adenosine triphosphate (ATP) in the pathophysiology of AD and assesses the therapeutic potential of clodronate, an ATP release inhibitor. Our research demonstrates that extracellular ATP impairs skin barrier function by reducing the filaggrin expression in the keratinocytes, a critical protein for barrier integrity. Furthermore, ATP release, triggered by IL-4 and mechanical stimuli, amplifies inflammation by promoting cytokine and chemokine production by the immune cells. Clodronate, by inhibiting ATP release, restores the filaggrin levels in the keratinocytes, reduces TARC production in the dendritic cells, and alleviates AD symptoms in a mouse model. These findings suggest that targeting extracellular ATP could offer a novel therapeutic approach to improving skin barrier function and reducing inflammation in AD. Future studies should explore the long-term efficacy and safety of ATP-targeted therapies in clinical settings.

## 1. Introduction

The skin, particularly the epidermis, plays a critical role in acting as a barrier between the body and the external environment, sensing various stimuli and maintaining homeostasis. However, abnormal sensing and responses to external stimuli can lead to aberrant immune responses and inflammation in the skin, contributing to various skin diseases [1]. Atopic dermatitis (AD) is a well-known inflammatory skin disorder characterized by abnormalities in skin barrier function, inflammation, and pruritus, which interact to form its pathophysiology [2]. Recently, many biologics-targeting cytokines that control inflammation and itching, as well as small-molecule compounds targeting intracellular cytokine signaling, have been developed and are being used clinically [3]. In AD, various external stimuli, such as contact with clothing and scratching due to itching, often exacerbate its symptoms [4,5].

Adenosine triphosphate (ATP) is known as the energy currency of the body and is involved in the storage and utilization of energy for life activities such as muscle contraction. When ATP is hydrolyzed by the action of ATPases, it loses one phosphate group to become adenosine diphosphate (ADP), releasing energy in the process, which is used for muscle contraction [6,7]. Several studies have reported that physical or chemical stimulation of the epidermal cells causes the release of ATP into the extracellular space [8,9]. Additionally, this released ATP acts on neighboring immune cells, promoting the production of cytokines and chemokines. This indicates that ATP-mediated intercellular interactions play a significant role in immunity, allergy, and inflammatory responses and are involved in various skin diseases, including AD [10,11,12,13,14,15,16,17,18,19,20,21]. Therefore, targeting ATP in therapeutic approaches, especially in the form of topical medications, may have the potential to improve AD symptoms. Recent research has revealed that clodronate, taken up by the nerve and immune cells, inhibits vesicular nucleotide transporter (VNUT) and selectively blocks ATP release, thus providing therapeutic effects for neuropathic and inflammatory pain [22].

In this study, we aimed to investigate the impact of ATP on the pathophysiology of AD and examine the potential therapeutic effects of clodronate on AD by inhibiting ATP, using cell and mouse models of AD.

## 2. Results

### 2.1. Inhibition of ATP Release Restores Barrier Dysfunction Caused by IL-4 or Scratching

Filaggrin expression is crucial for skin barrier function in AD. To examine the impact of ATP on skin barrier function, normal human epidermal keratinocytes (NHEKs) were stimulated with ATP, and the filaggrin expression was measured. ATP stimulation significantly reduced the filaggrin expression in the NHEKs (Figure 1A). To determine whether IL-4 stimulation, which is important in AD pathology, promotes ATP release by NHEKs, IL-4 was used to stimulate these cells. IL-4 stimulation promoted ATP release by the NHEKs, and clodronate significantly inhibited this release (Figure 1B). Subsequently, we confirmed whether clodronate could restore the reduction in filaggrin expression caused by IL-4 stimulation by inhibiting ATP release. As previously reported, IL-4 stimulation reduced the filaggrin expression in the NHEKs, and clodronate restored this reduced expression (Figure 1C). Next, scratching was applied to these cells to investigate whether mechanical stimuli (scratching) promote ATP release by NHEKs. Scratching promoted ATP release by the NHEKs, and clodronate significantly inhibited this release (Figure 1D). Consequently, we examined whether clodronate could restore the reduction in filaggrin expression caused by scratching stimulation by inhibiting ATP release. Scratching reduced the filaggrin expression in the NHEKs, and clodronate restored this reduced expression (Figure 1E).

### 2.2. Inhibition of ATP Release Suppresses IL-4-Induced TARC Production from Dendritic Cells

Next, we examined whether ATP promotes the production of the chemokine TARC, which is important in AD inflammation, by dendritic cells (DCs). When bone-marrow-derived DCs (BMDCs) were stimulated with ATP, TARC production from the BMDCs was promoted (Figure 2A,B). To investigate whether IL-4 stimulation promotes ATP release by BMDCs, IL-4 was used to stimulate these cells. IL-4 stimulation promoted ATP release by the BMDCs, and clodronate significantly inhibited this release (Figure 2C). Consequently, we confirmed whether clodronate could reduce the IL-4-induced promotion of TARC production by inhibiting ATP release. As previously reported, IL-4 stimulation promoted TARC production by the BMDCs, and clodronate inhibited this production (Figure 2D,E).

### 2.3. Topical Application of Clodronate Improves Symptoms in Mice with AD 

We examined whether topical application of clodronate was effective in mice with AD induced by repeated application of house dust mite antigens. Application of the house dust mites significantly increased the ear swelling in the mice, and topical application of clodronate significantly reduced this swelling (Figure 3A). Topical application of clodronate also significantly reduced clinical manifestations such as erythema, scale, and dryness (Figure 3B). Histopathological examinations confirmed this effect, showing that topical application of clodronate significantly reduced epidermal thickening and inflammatory cell infiltration in the ears (Figure 3C,D).

### 2.4. Topical Application of Clodronate Improves Inflammation and Barrier Dysfunction in Mice with AD

Subsequently, we investigated the effects of clodronate on the expression of various cytokines in the local skin. Repeated application of the house dust mites significantly increased the expression of many cytokines, including IL-4, IL-13, IFN-g, and TSLP, while clodronate application significantly reduced the expression of IL-4, IL-13, IFN- g, IL-17, and IL-33 (Figure 4A). We also confirmed the impact of clodronate on skin barrier function in the mice with AD. Application of the house dust mites increased the transepidermal water loss (TEWL), indicating skin barrier dysfunction, but topical application of clodronate significantly reduced the TEWL, improving skin barrier function (Figure 4B).

## 3. Discussion

In this study, we investigated the impact of extracellular ATP on the pathogenesis of AD and explored the potential therapeutic application of clodronate, a compound that inhibits ATP release. Our findings reveal significant insights into the role of ATP in skin barrier dysfunction and inflammatory processes, highlighting novel avenues for the treatment strategies in AD. 

Firstly, we observed that extracellular ATP markedly reduces filaggrin expression in keratinocytes. Filaggrin is a crucial protein for maintaining skin barrier integrity, and its deficiency is closely associated with the compromised barrier function observed in AD [23,24]. The reduction in filaggrin expression suggests that ATP plays a detrimental role in weakening the skin’s defensive barrier, thereby facilitating the penetration of allergens and irritants. This effect of ATP aligns with the broader understanding of how skin barrier dysfunction contributes to the chronic nature of AD, wherein the barrier’s impairment leads to a vicious cycle of inflammation and further barrier breakdown. Moreover, our study identified that the release of extracellular ATP is significantly enhanced by the presence of IL-4 and mechanical stimuli such as scratching, both of which are common in AD patients. This finding is particularly important, as it suggests a mechanistic link between the characteristic behaviors seen in AD, such as frequent scratching, and the exacerbation of skin inflammation through increased ATP release. The observation that IL-4, a cytokine heavily involved in the Th2 immune response, can induce ATP release further cements the role of ATP in the inflammatory cascade associated with AD. This connection between cytokine signaling, physical stimuli, and ATP release offers a more comprehensive understanding of the factors that drive the persistence and severity of AD symptoms.

In addition to elucidating the pathogenic role of ATP, our study explored the therapeutic potential of clodronate, a bisphosphonate known to inhibit ATP release [25]. The application of clodronate resulted in the restoration of filaggrin expression and a significant reduction in TARC production in murine BMDCs. TARC is a key mediator in the recruitment of Th2 cells to sites of inflammation [26,27], and its suppression indicates that clodronate may effectively reduce the Th2-driven inflammatory response in AD. This result is promising, as it suggests that clodronate, or similar agents targeting ATP, could be developed as topical treatments to ameliorate the symptoms of AD by not only improving skin barrier function but also by dampening the underlying inflammatory processes.

Finally, our in vivo experiments using an AD mouse model provided further evidence of clodronate’s efficacy. Mice treated with clodronate exhibited significant improvements in their AD symptoms, including reduced ear swelling, decreased inflammatory cell infiltration, and a better overall condition of the skin. These findings confirm the potential of clodronate as a therapeutic agent that could be particularly useful in managing the acute exacerbations of AD that are triggered by environmental factors or stress.

AD symptoms often worsen suddenly due to various external stimuli in addition to persistent symptoms [28]. The current AD treatments mainly aim to suppress persistent symptoms. Since ATP is rapidly released extracellularly in response to external stimuli, it is suggested that ATP may be involved in the sudden worsening of AD symptoms. Thus, ATP-targeted therapy may be effective in reducing symptoms of acute exacerbations, as well as ongoing chronic symptoms. In this study, the therapeutic efficacy of clodronate, which inhibits ATP release, was tested as an example of the potential of ATP inhibition in the treatment of AD. In the future, the search for substances that inhibit ATP through various mechanisms may lead to the development of new AD drugs. It is very reasonable that keratinocytes, the outermost layer of the skin in contact with the outside world, release ATP upon external stimulation, which induces nociceptive responses, i.e., pain and itching, as biological defense responses. The potential of targeting ATP release as a treatment for pain and itching has already been reported [11,22], and its application in topical therapy warrants further investigation. In fact, clodronate, which we used in this study, has demonstrated potent and reversible analgesic effects in models of neuropathic and inflammatory pain in mice. VNUT-deficient mice exhibited reduced sensitivity to these types of pain, and the analgesic effect of clodronate was absent, suggesting that clodronate exerts its analgesic effects by targeting VNUT. Additionally, clodronate has been shown to inhibit VNUT, thereby suppressing ATP release from nerve cells, suggesting that the mechanism of clodronate’s analgesic action involves the inhibition of ATP release via VNUT suppression [22]. Therefore, although it was not examined in this study, clodronate may have therapeutic potential for targeting the itching and pain of AD.

In conclusion, this study underscores the critical role of extracellular ATP in the pathophysiology of AD, particularly concerning skin barrier dysfunction and inflammatory amplification. The inhibition of ATP release by clodronate presents a novel therapeutic approach that warrants further investigation, especially in clinical settings (Figure 5). Future research should focus on exploring the long-term effects of clodronate treatment in AD patients and determining its efficacy across different stages of this disease. Additionally, investigating the potential synergistic effects of clodronate with other AD treatments could lead to more effective and comprehensive management strategies for this chronic and debilitating condition.

## 4. Materials and Methods

### 4.1. In Vitro Experiments

#### 4.1.1. Cell Culture

The NHEKs were purchased from Lonza (Basel, Switzerland) and cultured in KGM-Gold (Lonza) supplemented with bovine pituitary extract, recombinant human epidermal growth factor, insulin, hydrocortisone, gentamycin–amphotericin, transferrin, and epinephrine at 37 °C in 5% CO_2_. The medium was changed every 2 days. The cells reached 70–80% confluence and were passaged three times. The third passage of cells was then treated with ATP (Sigma-Aldrich, MO, USA) or IL-4 (Peprotech, Rocky Hill, NJ, USA) with or without pretreatment with clodronate (Sigma-Aldrich, St. Louis, MO, USA).

#### 4.1.2. Generation of the BMDCs and Cell Culture

C57BL/6 mice were housed in a clean facility and bred and used under the guidelines of the animal facility center of Kyushu University. Bone marrow (BM) cells freshly isolated from the mice were cultured in RPMI-1640 medium supplemented with 10% FBS (CAPRICORN Scientific, Ebsdorfergrund, Germany), 10 mmol/L 4-(2-hydroxyethyl)-1-piperazineethanesulfonic acid (HEPES) (Thermo Fisher Scientific, Waltham, MA, USA), 1% Minimum Essential Medium Non-Essential Amino Acids (MEM NEAAs) (Thermo Fisher Scientific), 1 mmol/L sodium pyruvate (Thermo Fisher Scientific), Antibiotic–Antimycotic (100 U/mL penicillin, 100 mg/mL streptomycin, and 0.25 µg/mL amphotericin B) (Thermo Fisher Scientific), and 4 ppm β-mercaptoethanol (Nacalai tesque, Kyoto, Japan) in the presence of 10 ng/mL GM-CSF (Peprotech). The medium was refreshed and GM-CSF was added twice in 8 days. On day 10, non-adherent cells were harvested. These cells were purified immunomagnetically through two or three rounds of positive selection with CD11c MicroBeads (Miltenyi Biotec, North Rhine-Westphalia, Bergisch Gladbach, Germany). As we described previously [29], the purity of the DCs, as determined by fluorescence-activated cell sorting (FACS), was between 95% and 97%. The purified BMDCs were cultured with IL-4 (Peprotech, Rocky Hill, NJ, USA), clodronate, and ATP. The culture supernatants were collected at 24 h. Cells were also collected for qRT-PCR analysis.

#### 4.1.3. The In Vitro Scratched Keratinocyte Model

To establish the in vitro scratched keratinocyte model [30], NHEKs were seeded into 6-well plates (4 × 10^5^ cells/well). Entire confluent keratinocyte sheets were scratched with 3, 9, and 18 lines, using 1000 µL tips, and incubated for 15 min for the measurement of extracellular ATP or for 24 h for q-RT PCR at 37 °C in 5% CO_2_ after scratching.

#### 4.1.4. Measurement of Extracellular ATP

The extracellular ATP concentration was measured using ENLITEN^®^ rLuciferase/Luciferin Reagent (Promega, Madison, WI, USA). To investigate scratching- or IL-4-induced ATP release, conditioned medium was collected as a control sample to measure the background ATP release. At 15 min after scratching or IL-4 stimulation, each sample was centrifuged at 400 × g for 5 min, and 250 µL of the supernatant was collected for ATP determination. The concentration of ATP was determined by adding 100 µL of luciferin–luciferase reagent to 100 µL of the sample solution, followed by measuring the chemiluminescence using a DTX 800 Multimode Detector (Beckman Coulter Inc., Brea, CA, USA).

#### 4.1.5. Reverse Transcription and Quantitative Real-Time PCR (qRT-PCR)

Total RNA was extracted from the cells using the RNeasy Plus Micro Kit (Qiagen, Hilden, Germany), and cDNA was synthesized using the PrimeScript RT reagent kit (Takara Bio, Shiga, Japan). qRT-PCR was performed on the CFX Connect Real-Time PCR Detection System (Bio-Rad, Hercules, CA, USA) using TB Green Premix Ex Taq (Takara Bio). Denaturation was set at 95 °C for 30 s with 40 total cycles and with a second step at 95 °C for 5 s. Annealing occurred at 60 °C for 20 s. The sequences of primer pairs are shown in Appendix A. 

#### 4.1.6. Enzyme-Linked Immunosorbent Assay (ELISA)

Each culture supernatant was measured using murine CCL17/TARC ELISA kits (R&D Systems, Inc., Minneapolis, MN, USA) in accordance with the manufacturer’s protocols. The optical density was measured using a DTX 800 Multimode Detector (Beckman Coulter Inc., Brea, CA, USA).

### 4.2. In Vivo Animal Experiments

#### 4.2.1. Reagents

Clodronate (dichloromethylene diphosphonic acid disodium salt), purchased from Sigma-Aldrich (St. Louis, MO, USA), was dissolved in Dulbecco’s Phosphate-Buffered Saline (D-PBS (-)); Nacalai Tesque, Kyoto, Japan), with the pH of the solutions adjusted to 7 with NaOH, and stored at 4 °C until its use for the mouse experiments. MC903, purchased from Tocris (Avonmouth, Bristol, UK), was dissolved in ethanol (Nacalai Tesque, Kyoto, Japan) and stored at −30 °C until its use for the mouse experiments.

#### 4.2.2. Mice

Female C57BL/6 mice aged 8 weeks were purchased from Japan SLC, Inc. (Shizuoka, Japan). The mice were maintained under a 12 h light/dark cycle under specific pathogen-free conditions. The animal experiments in this study were approved by the Animal Care and Use Committee of Kyushu University School of Medicine.

#### 4.2.3. The Murine Model of Mite-Antigen-Induced AD 

The mice were divided into the following three groups—the control group, the AD group, and the AD + clodronate group—on day 0. In the AD and the AD + clodronate groups, both ears were painted with 2.5 nmol of MC903 in 25 µL ethanol. Five hours later in the AD + clodronate group, both ears were topically treated with 20 µL of 50 mg/mL clodronate. In the control group, ethanol was applied to replace MC903 as the stimulant. After 5 consecutive days, the ears continued to receive clodronate or the vehicle daily without MC903 until day 10. Ear thickness was measured daily using a digital caliper (Mitutoyo Corp., Tokyo, Japan).

#### 4.2.4. Histological Examination

One of each mouse’s ears was fixed in 10% neutral formalin, embedded into paraffin, and sectioned into 3 µm slices. The sections were deparaffinized, rehydrated, and stained with hematoxylin and eosin. To assess the number of inflammatory cells in the dermis, three high-magnification fields per skin section from each mouse were randomly selected, and the average number of stained cells was counted. In measuring the epidermal thickness, measurements were taken at five random locations with a low magnification field of view and averaged.

#### 4.2.5. Quantitative Reverse Transcription PCR (qRT-PCR) Analysis

RNA was obtained from the ear skin by homogenizing it with TRIzol^®^ reagent (Thermo Fisher Scientific, Waltham, MA, USA) and the RNeasy Plus Mini kit (Qiagen, Hilden, Germany) with the RNase-Free DNase Set (Qiagen) according to the manufacturers’ instructions. Reverse transcription was performed using the PrimeScript RT reagent kit (Takara Bio, Otsu, Japan). qRT-PCR was conducted on a CFX Connect Real-Time System (Bio-Rad, Hercules, CA, USA) using TB Green Premix Ex Taq (Takara Bio, Shiga, Japan). Amplification was initiated at 95 °C for 30 s as the first step, followed by 40 cycles of 95 °C for 5 s and at 60 °C for 20 s. The mRNA expression was measured in duplicate and normalized to the Tbp expression level. The sequences of primer pairs are shown in Appendix A.

#### 4.2.6. Transepidermal Water Loss (TEWL)

The TEWL was measured in the skin lesions using a Vapo Scan AS-VT100RS machine (ASCH Japan Co., Ltd., Tokyo, Japan). Measurement of the TEWL was performed at room temperature (23–26 °C) and in room humidity (40–55%) in the animal facility center of Kyushu University.

#### 4.2.7. Statistical Analysis

All the data are presented as means ± standard errors of the mean (SEMs). The significance of the differences between groups was assessed using one-way ANOVA, followed by Bonferroni’s multiple comparison test, using GraphPad PRISM 5 software version 5.02 (GraphPad Software, La Jolla, CA, USA). A *p*-value of less than 0.05 was considered statistically significant.

## Figures and Tables

**Figure 1 ijms-25-12294-f001:**
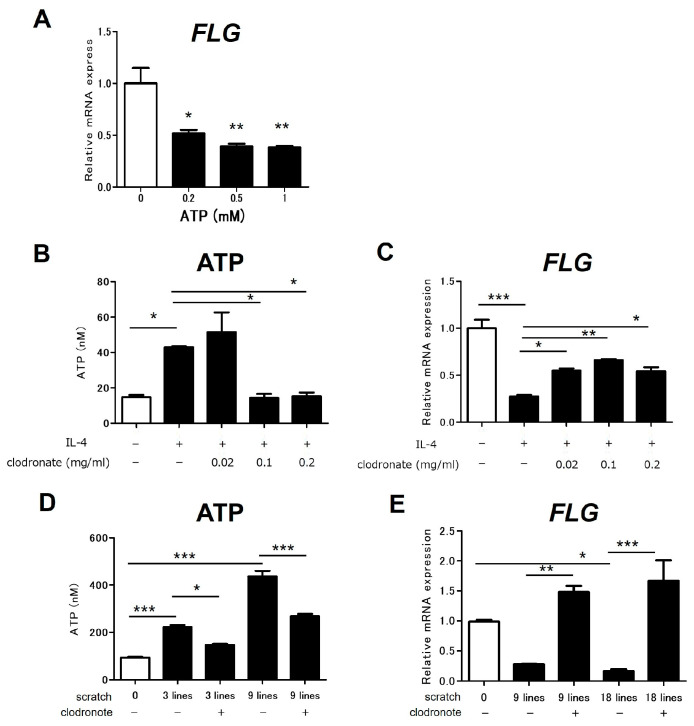
Inhibition of ATP release restored the reduced expression of filaggrin induced by IL-4 and scratching in NHEKs. NHEKs were stimulated with ATP (**A**), IL-4 (**B**,**C**), or scratching (**D**,**E**) in the presence or absence of clodronate. The expression levels of FLG mRNA (**A**,**C**,**E**) were analyzed using qRT-PCR. The extracellular ATP concentration was measured using a luciferin–luciferase assay (**B**,**D**). All data are presented as means ± SEM. The results are representative of similar results obtained in three independent experiments. * *p* < 0.05, ** *p* < 0.01 and *** *p* < 0.001; one-way ANOVA, followed by Bonferroni’s multiple comparison test.

**Figure 2 ijms-25-12294-f002:**
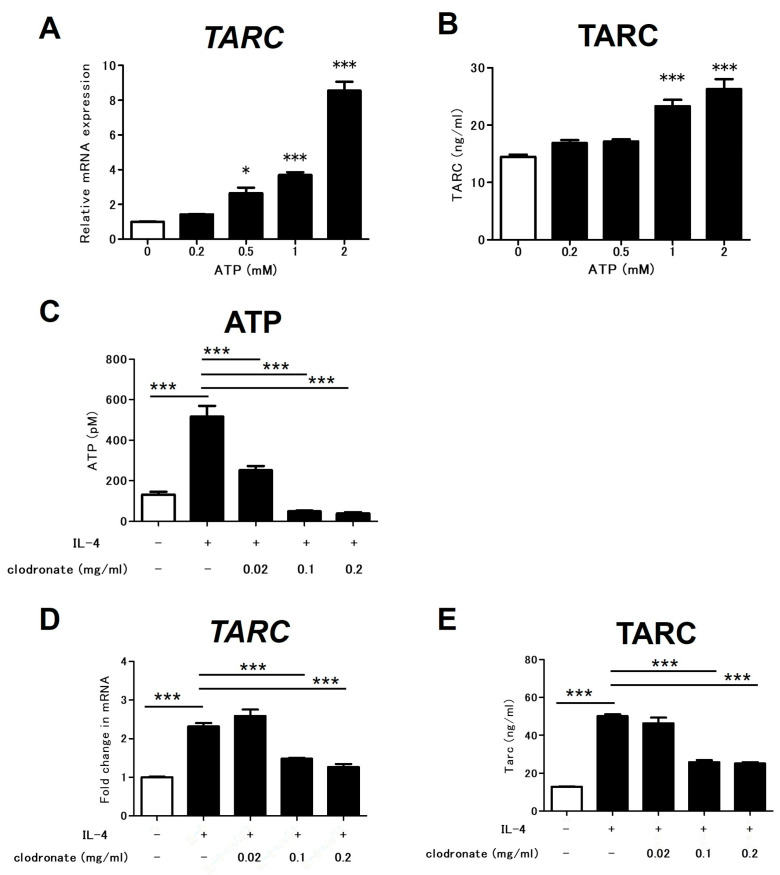
Inhibition of ATP release suppresses IL-4-induced TARC production by BMDCs. BMDCs were treated with ATP (**A**,**B**). TARC mRNA expression was analyzed using qRT-PCR (**A**), and TARC protein production was analyzed using ELISA (**B**). BMDCs were treated with IL-4 in presence or absence of clodronate (**C**–**E**). Extracellular ATP concentration was measured using luciferin–luciferase assay (**C**). TARC mRNA expression was analyzed using qRT-PCR (**D**), and TARC protein production was analyzed using ELISA (**E**). All data are presented as means ± SEM. Results are representative of similar results obtained in three independent experiments. * *p* < 0.05, *** *p* < 0.001; one-way ANOVA, followed by Bonferroni’s multiple comparison test.

**Figure 3 ijms-25-12294-f003:**
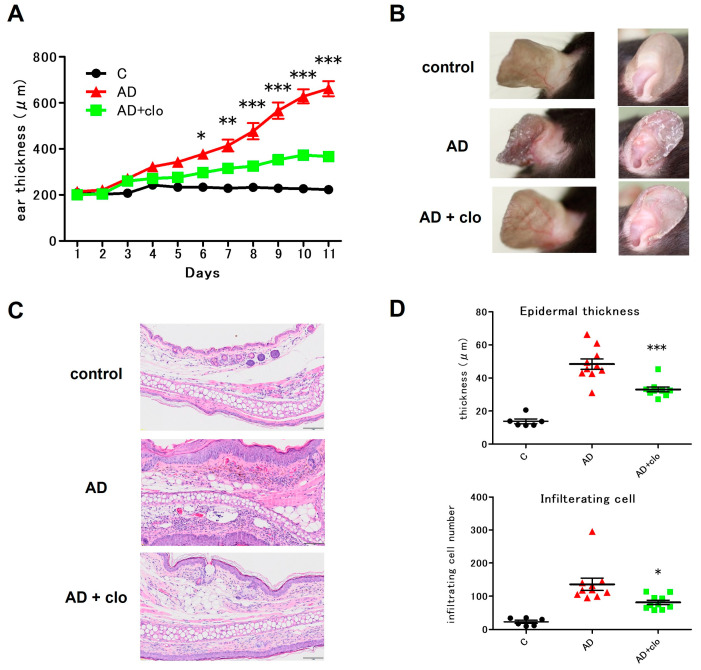
Topical application of clodronate improves the clinical symptoms of and histological changes in mite-induced AD in NC/Nga mice. The mice were divided into the following three groups—a control group, an AD group, and an AD + clodronate group—on day 0. In the AD and the AD + clodronate groups, both ears were painted with MC903. Five hours later in the AD + clodronate group, both ears were topically treated with clodronate (in the AD + clodronate group) or vehicle (in the AD group). In the control group, ethanol was applied to replace MC903 as the stimulant. After 5 consecutive days, the ears continued to receive clodronate and ethanol daily without MC903 until day 10. Ear thickness was measured daily using a digital caliper. (**A**) Ear thickness was measured at every mite antigen stimulation. (**B**) Macroscopic features of skin lesions in control, AD, and AD + clodronate groups on days 0 and 6. (**C**) Histological appearance (hematoxylin-eosin staining) of skin lesions in the control, AD, and AD + clodronate groups on day 7. Scale bar: 50 μm. (**D**) Measurements of epidermal thickness and numbers of total inflammatory cells in the dermis of the ear skin of the mice. All data are presented as means ± SEM. (*n* = 10 per group). The results are representative of similar results obtained in three independent experiments. * *p* < 0.05, ** *p* < 0.01, and *** *p* < 0.001; one-way ANOVA, followed by Bonferroni’s multiple comparison test.

**Figure 4 ijms-25-12294-f004:**
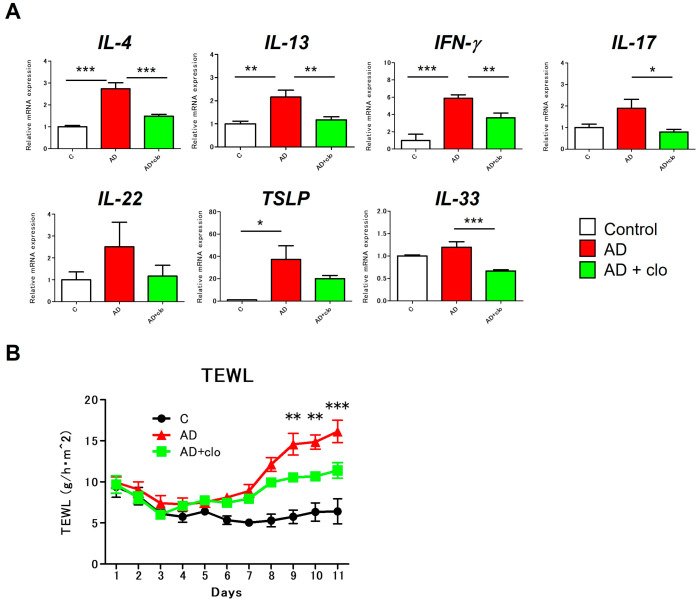
Topical application of clodronate improves cytokine expression and the TEWL in AD mice. (**A**) mRNA levels of the cytokines were analyzed using real-time PCR in the skin of the NC/Nga mice. (**B**) The TEWL was measured in a lesion using a Vapo Scan instrument on days 0, 2, 4, and 6. Data are presented as the average of repeated recordings at five points. All data are presented as means ± SEM. (*n* = 6 control, *n* = 10 AD, *n* = 10 AD + clodronate). Results are representative of similar results obtained in three independent experiments. * *p* < 0.05, ** *p* < 0.01, and *** *p* < 0.001.

**Figure 5 ijms-25-12294-f005:**
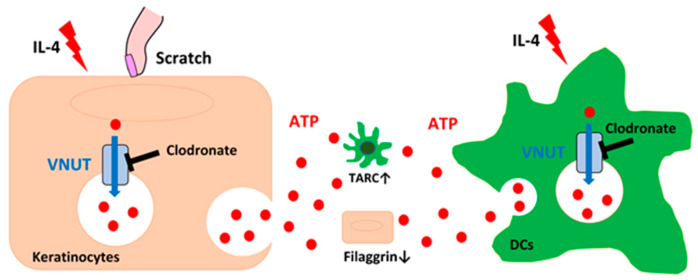
The release and effects of ATP in response to external stimulation of keratinocytes and DCs in the skin. Keratinocytes in the outermost skin layer are shown to release ATP upon stimulation (e.g., scratching or IL-4 exposure). The release mechanism involves VNUT, vesicular nucleotide transporter, which clodronate can inhibit to reduce ATP release. DCs similarly release ATP in response to stimulation, with clodronate again targeting VNUT to inhibit ATP release. In the extracellular space, ATP interacts with surrounding cells, such as keratinocytes and DCs. This ATP signaling increases the expression of TARC, a chemokine involved in immune cell recruitment, and decreases filaggrin levels, a key protein for skin barrier function.

## Data Availability

The authors confirm that the data supporting the findings of this study are available within the article and its Appendix A.

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
