# Peer review of "Extracellular ATP Contributes to Barrier Function and Inflammation in Atopic Dermatitis: Potential for Topical Treatment of Atopic Dermatitis by Targeting Extracellular ATP"

_ijms, 2024, doi:10.3390/ijms252212294_

Round 1
Reviewer 1 Report
Comments and Suggestions for Authors
Dear authors,
Congratulations on your valuable work. It was a pleasure reading your manuscript. The research focuses on a topic of great interest, its novelty is indisputable and the importance of the results for general clinical practice are evident.
I have a few suggestions:
1. The manuscript would benefit from a more thorough description of the ability of keratinocytes to initiate nociceptive responses by the release of microvesicles containing ATP following exposure to various stimuli.
2. A diagram illustrating the extracellular release of ATP would be helpful.
3. Please consider discussing clodronate’s mechanism of action and analgesic properties in more detail.
4. The interesting and complex mutual influence of IL-4 and extracellular ATP should also be presented.
Best regards!
Author Response
Dear Reviewer 1,
Thank you for taking the time to review our paper. Thank you also for your encouraging comments.
- The manuscript would benefit from a more thorough description of the ability of keratinocytes to initiate nociceptive responses by the release of microvesicles containing ATP following exposure to various stimuli.
Thank you for your very important suggestion. I am adding the following to the discussion.
It is very reasonable that keratinocytes, the outermost layer of skin in contact with the outside world, release ATP upon external stimulation, which induces nociceptive responses, i.e., pain and itch as biological defense responses. The potential of targeting ATP release as a treatment for pain and itching has already been reported, and its application in topical therapy warrants further investigation.
- A diagram illustrating the extracellular release of ATP would be helpful.
Thank you very much for your suggestion.
I have added a diagram as Figure 5.
- Please consider discussing clodronate’s mechanism of action and analgesic properties in more detail.
Thank you for your comment.
I have added a more detailed mechanism of analgesic action as follows.
 In fact, clodronate, which we used in this study, has demonstrated potent and reversible analgesic effects in models of neuropathic and inflammatory pain in mice. VNUT-deficient mice exhibited reduced sensitivity to these types of pain, and the analgesic effect of clodronate was absent, suggesting that clodronate exerts its analgesic effects by targeting VNUT. Additionally, clodronate has been shown to inhibit VNUT, thereby suppressing ATP release from nerve cells, suggesting that the mechanism of clodronate's analgesic action involves the inhibition of ATP release via VNUT suppression. Therefore, although not examined in this study, clodronate may have therapeutic potential for targeting the itching and pain of AD.
- The interesting and complex mutual influence of IL-4 and extracellular ATP should also be presented.
Thank you for your important comment.
It is possible that IL-4 and ATP are both localized and synergistic in the AD skin localization due to the chronic course of the disease. We have not been able to verify this in this case, but we have included this possibility in our discussion.
Again, thank you for your valuable comments. Thanks to you, I am convinced that the quality of the paper has improved.
Reviewer 2 Report
Comments and Suggestions for Authors
It is my great pleasure to have an opportunity to review this interesting article. The authors investigated the contribution of ATP to skin barrier function and inflammation in AD and discussed the potential therapeutic efficacy of topical anti-ATP agents. This article is fascinating and well-written. I have some comments and questions below.
1. In Figure 1B, 0.02 mg/mL of clodronate didn’t reduce ATP release. However, 0.02 mg/mL of clodronate restored the IL-4-induced filaggrin reduction significantly. What do you think it means? Does clodronate have any function to restore filaggrin expression other than inhibition of ATP release?
2. In Figure 1E, clodronate increased filaggrin expression more than the basal expression level. Does clodronate have any function not only to restore filaggrin expression from scratch-induced reduction but also to increase filaggrin expression compared to basal level?
3. In Figures 3 and 4, did you check the filaggrin expression in the topical application mouse model? To discuss the barrier function, local filaggrin expression should be presented.
4. In Figures 3 and 4, did topical use of clodronate decrease serum TARC level in this mouse model? It is fascinating that topical clodronate application improves cytokine expression in AD mice. To discuss the mechanism of amplification of inflammation, the TARC level should be presented.
Author Response
Dear Reviewer 2
Thank you for taking the time to review our paper. We also appreciate your valuable comments.
- In Figure 1B, 0.02 mg/mL of clodronate didn’t reduce ATP release. However, 0.02 mg/mL of clodronate restored the IL-4-induced filaggrin reduction significantly. What do you think it means? Does clodronate have any function to restore filaggrin expression other than inhibition of ATP release?
Thank you for your very important point.
The results of clodronate 0.02 mg/mL inhibition of ATP release were varied and difficult to interpret. This may be due to the experimental conditions. We would like to study the effects of clodronate other than ATP release inhibition in the future. At the same time, to examine the importance of ATP inhibition, we would like to consider examining the difference in effect between clodronate and other ATP release inhibitors.
- In Figure 1E, clodronate increased filaggrin expression more than the basal expression level. Does clodronate have any function not only to restore filaggrin expression from scratch-induced reduction but also to increase filaggrin expression compared to basal level?
Thank you for your important comment.
Figure 1E did not show a statistically significant increase. However, it may be possible that clodronate increases normal filaggrin expression without IL-4 or scratching.
- In Figures 3 and 4, did you check the filaggrin expression in the topical application mouse model? To discuss the barrier function, local filaggrin expression should be presented.
This is a very important point.
We are now planning to examine the effects of clodronate in vivo in more detail using various mouse models of AD. We plan to examine the expression of barrier-related proteins in skin.
- In Figures 3 and 4, did topical use of clodronate decrease serum TARC level in this mouse model? It is fascinating that topical clodronate application improves cytokine expression in AD mice. To discuss the mechanism of amplification of inflammation, the TARC level should be presented.
This is a very important point.
We are now planning to examine the effects of clodronate in vivo in more detail using various mouse models of AD in this regard as well. We also plan to examine biomarkers in blood such as TARC, IgE.
Although we were not able to add further data to this paper, you have given us some very important insights for our future work. Again, thank you very much. We would like to continue our research to clarify the answers to your comments in this issue.